# Clinical presentation of Oropouche virus infection: A systematic review and meta-analysis

Zhilin Wang[1,2☯], Linzhu Huang[1☯], Xinyu Zhang[1], Xinyue Zhang[1], Liwei Huang[1], Xiaoying Zhu[1,3,4], Xidai Long[1,3,4]\*, Demin Cao[1,3,4]\*, Yulei Li[1,3,4]\*

**1** Youjiang Medical University for Nationalities, Baise, China, **2** Jiangxi Provincial Key Laboratory of Conservation Biology, College of Forestry, Jiangxi Agricultural University, Nanchang, China, **3** Clinicopathological Diagnosis & Research Center, the Affiliated Hospital of Youjiang Medical University for Nationalities, Baise, China, **4** Key Laboratory of Tumor Molecular Pathology of Guangxi Higher Education Institutes, Baise, China,

☯ These authors Contributed equally to this work.
\* liyulei@ymun.edu.cn (YL); caodm@ymun.edu.cn (DC); sjtulongxd@ymun.edu.cn (XL)

## Abstract

### Background

The recent surge in incidence and geographic spread of OROV infections poses an escalating threat to global public health. However, studies exploring the clinical signs of OROV infection remains exceedingly limited.

### Methods

We searched for OROV studies published until June 17, 2024, in several electronic databases including MEDLINE, EMBASE, SCOPUS, and the Cochrane Library.

### Results

In total, 15 studies involving 806 patients with OROV infection were eligible for inclusion. General symptoms with fever and headache were the most common. Gastrointestinal disturbances like nausea/vomiting, anorexia, and odynophagia were also prevalent, along with ocular symptoms, mainly retro-orbital pain, photophobia, and redness. Respiratory symptoms, such as cough, sore throat and nasal congestion, are present, and skin-related issues like rash, pruritus, and pallor were also identified.

### Conclusion

Overall, this study provides a foundational understanding of OROV's clinical manifestations to guide diagnosis, management, and public health interventions against this neglected tropical disease.

**Data availability statement:** All relevant data are in the manuscript and its supporting information files.

**Funding:** This study was supported in by Joint Special project of Guangxi Natural Science Foundation (2025GXNSFHA069143 to Y.L.;2025GXNSFHA069078 to D.C.). The funders had no role in study design, data collection, and interpretation, or submitting the work for publication.

**Competing interests:** The authors have declared that no competing interests exist.

## Author summary

In the realm of life sciences, understanding the full scope of infectious diseases is crucial for protecting public health. Oropouche virus (OROV), a relatively under-studied pathogen, has been showing an alarming increase in both incidence and geographical spread recently. Despite its growing threat, our knowledge of the clinical symptoms it causes has been severely lacking. Our study is the first of its kind to comprehensively review and analyze available research on OROV-related symptoms. By pooling 15 studies involving 806 patients, we've uncovered a range of symptoms from common fever and headache to less-known ocular, gastrointestinal, and skin - related issues. This new understanding is vital. For scientists, it lays the groundwork for further research into OROV's biology and disease mechanisms. For non-scientists, it helps in early recognition of the disease, which is key to getting proper medical care and preventing its spread.

## Introduction

Oropouche virus (OROV) is an arbovirus member of the *Peribunyaviridae* family and the *Orthobunyavirus* genus. Since its first discovery in 1955 in Trinidad and Tobago [1], OROV has recurrently provoked epidemics throughout South and Central American territories [2]. These episodic outbreaks are often linked to midge vector (*Culicoides paraensis*) with sloths and non-human primates serving as reservoir hosts, primarily occurred in the urban settings of Amazon basin region (like the Brazilian Amazon), where there is often a proliferation of insects [3,4]. However, in recent years, a notable increase in the incidence and geographical spread of reported OROV infections has been observed, highlighting a growing public health concern [5,6]. On February 2, 2024, the Pan American Health Organization (PAHO, WHO) issued an alert about a surge in OROV infection cases in some American countries in recent months [7], sparking concerns that the virus could be the next one to cause a major outbreak in Latin America.

OROV causes an acute febrile disease with diverse clinical manifestations. It often overlaps with common febrile diseases in regions such as dengue fever, Zika virus disease and chikungunya, which brings great challenges to clinical differential diagnosis [8–10]. To date, the comprehensive picture of clinical manifestations resulting from OROV infections remain poorly understood and characterized, as there is an evident lack of systematic reviews. This systematic review aims to summarize available evidence regarding the clinical presentation in OROV infections from available reports and studies.

## Methods

### Search strategy and data sources

A comprehensive search was performed in several electronic databases including MEDLINE, EMBASE, SCOPUS, and the Cochrane Central Register of Controlled Trials. This systematic search encompassed peer-reviewed studies to ensure the retrieval of high-quality and validated research related to the topic of interest. In addition, the reference lists of the included studies were screened for potentially eligible studies. We used controlled vocabulary supplemented with keywords search terms related to Oropouche virus. Only articles published in English were included in this systematic review. Studies published until June 17, 2024, were included. This meta-analysis study was not registered.

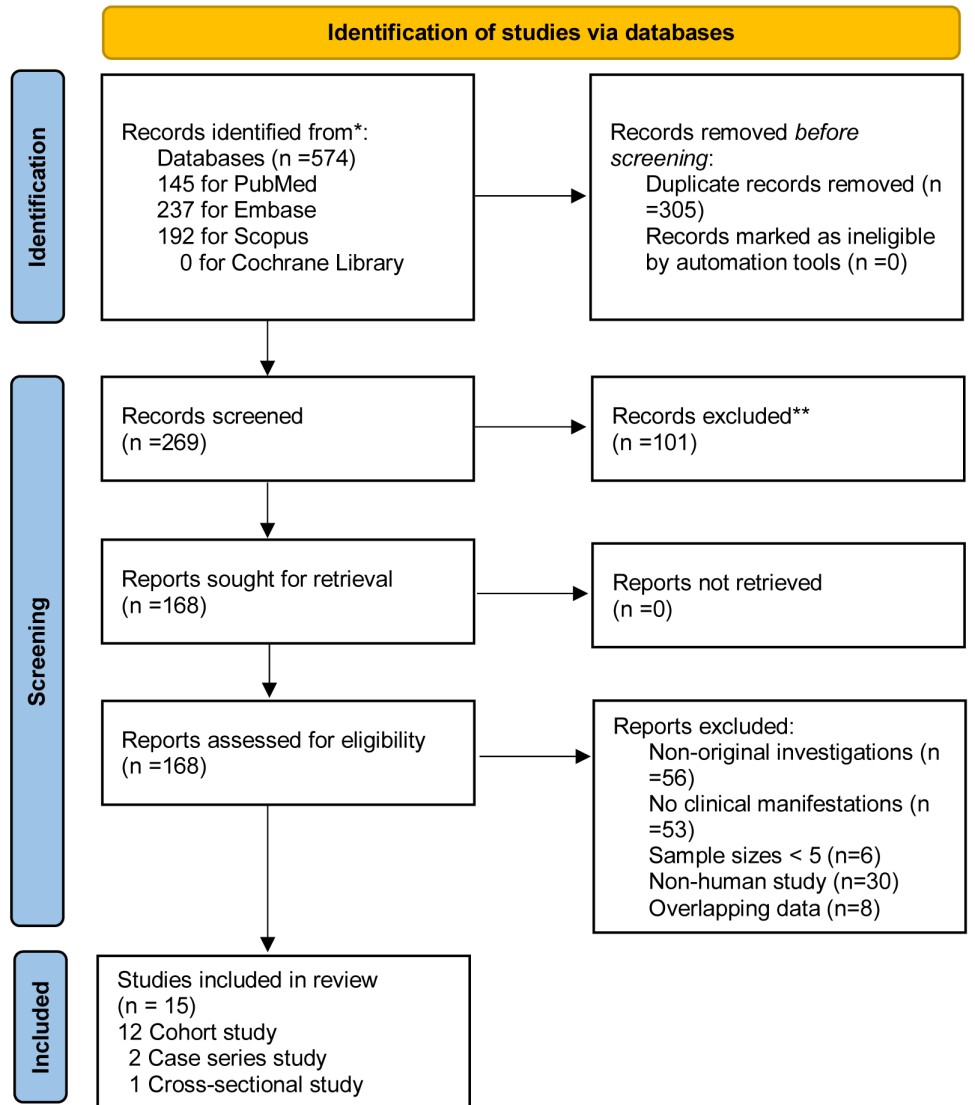

**Fig 1. Preferred Reporting Items for Systematic Reviews and Meta-Analyses (PRISMA) flowchart—showing the literature search strategy and the numbers of peer-reviewed articles included in, and excluded from, this study.**

## Eligibility criteria

Eligible studies had to meet all of the following inclusion criteria: (1) OROV infections were confirmed by polymerase chain reaction (PCR)or enzyme-linked immunosorbent assay (ELISA), and (2) studies containing detailed descriptions of clinical manifestations. Any observational studies (cohort, case-control, cross-sectional or case series) are included, no matter prospective or retrospective. Exclusion criteria were as follows: (1) Studies with sample sizes smaller than five, (2) studies on non-human hosts, (3) non-original investigations: reviews, meta-analyses, letters, commentaries, editorials, errata, and other articles.

## Study selection and data extraction

We used a two-stage screening approach: first by title and abstract and then by full text article. Two researchers independently screened each title, abstract, and full text and resolved

**Table 1. Baseline characteristics of OROV-infected patients included in the study.**

| No | Study | N | Study design | Country/region | Age, years mean±SD/mdeian(IQR) | Sex (man) | Specimen detection | Study period |
|----|-------|---|--------------|----------------|-------------------------------|-----------|--------------------|--------------|
| 1 | Aguilar et al. 2011 [14] | 16 | Cohort | Peru | NA | NA | Viral culture; PCR | 2003-2006 |
| 2 | Alva-Urcia et al. 2017 [15] | 12 | Cross-sectional | Peru | NA(range 5 to >45 years) | 8 | PCR | January-March 2016 |
| 3 | Azevedo et al. 2007 [16] | 93 | Cohort | Brazil | NA(range 1 to >55 years) | 35 | HI; ELISA | 2003-2004 |
| 4 | Cardoso et al. 2015 [17] | 5 | Cohort | Brazil | NA(range 14 to 62 years) | 1 | RT-qPCR | October 2011-July 2012 |
| 5 | Carvalho et al. 2022 [18] | 39 | Cohort | Brazil | NA | NA | Viral culture; ELISA | January-February 2018 |
| 6 | Ciuoderis et al. 2022 [19] | 105 | Cohort | Colombian | NA(range <18 to >18 years) | 55 | RT-qPCR | 2019-2022 |
| 7 | da Costa et al. 2017 [20] | 6 | Cohort | Brazil | 29.5±7 | NA | ELISA | 2011–2013 |
| 8 | Gaillet et al. 2021 [21] | 23 | Case series | French Guiana | NA(range <18 to >18 years) | 14 | PCR | August–September 2020 |
| 9 | Martins-Luna et al.2020 [22] | 131 | Cohort | Peru | NA(range <5 to >5 years) | 72 | PCR | February-September 2016 |
| 10 | Moreira et al. 2024 [23] | 12 | Cohort | Brazil | 43±17.68 | 16 | RT-qPCR | January 2022-March 2023 |
| 11 | Mourão et al. 2009 [24] | 128 | Cohort | Brazil | 29.5±14 | 51 | ELISA | January 2007-November 2008 |
| 12 | Naveca et al. 2018 [25] | 9 | Case series | Brazil | 36.2(range 8 to 63 years) | 7 | PCR | April-June 2015 |
| 13 | Silva-Caso et al. 2019 [26] | 46 | Cohort | Peru | 22.3±15.6 | 25 | PCR | January-July 2016 |
| 14 | Vasconcelos et al. 2009 [27] | 113 | Cohort | Brazil | NA(range 0 to >55 years) | 49 | HI; ELISA; Viral culture; PCR | May-June 2006 |
| 15 | Watts et al. 2022 [8] | 68 | Cohort | Peru | 31(range 1 to >44 years) | 44 | Viral culture; ELISA | 1993-1999 |

Abbreviations: ELISA, enzyme linked immunosorbent assay; HI, hemagglutination inhibition; NA, not applicable; PCR, polymerase chain reaction; RT-qPCR, quantitative reverse transcription polymerase chain reaction; SD, standard deviation.

discrepancies by consensus with a third researcher. For studies that have met the inclusion criteria, the following data were extracted using an extraction form: (1) author and its publication date, (2) study design, (3) Study period, (4) location of study, (5) baseline characteristics: sample size, age, sex, specimen detection, (6) clinical presentation.

## Quality assessment

The risk of bias of each study was independently evaluated by two authors using the methodological quality and synthesis of case series and case reports described by Murad et al [11]. All discrepancies were resolved by the independent opinion of a third reviewer. This article followed the Preferred Reporting Items for Systematic Reviews and Meta-Analyses (PRISMA) (S1 Checklist).

## Data analysis

We conducted this meta-analysis on R (version 4.0.3) using the meta and metafor packages. The fixed effect model will be used to derive the pooled prevalence estimates. If the prevalence in a study was reported as zero, we added a continuity correction of 0.5 to the number of cases before statistical analysis [12,13]. Forest plot was used to visualize the proportion of clinical presentation in OROV infections in each study and the overall pooled prevalence estimates. The $I^2$ statistic was used to estimate the level of heterogeneity. An $I^2$ of >75% indicated

**Table 2.  Methodological quality assessment tool results.**

| Study | Question 1 | Question 2 | Question 3 | Question 4 | Question 5 |
|---|---|---|---|---|---|
| Aguilar et al. 2011 [14] | Yes | Yes | Yes | Yes | Yes |
| Alva-Urcia et al. 2017 [15] | Yes | Yes | Yes | Yes | Yes |
| Azevedo et al. 2007 [16] | Yes | Yes | Yes | Yes | Yes |
| Cardoso et al. 2015 [17] | Yes | Yes | Yes | Yes | Yes |
| Carvalho et al. 2022 [18] | Yes | Yes | Yes | No | Yes |
| Ciuoderis et al. 2022 [19] | Yes | Yes | Yes | Yes | Yes |
| da Costa et al. 2017 [20] | Yes | Yes | Yes | Yes | Yes |
| Gaillet et al. 2021 [21] | Yes | Yes | Yes | Yes | Yes |
| Martins-Luna et al.2020 [22] | Yes | Yes | Yes | Yes | Yes |
| Moreira et al. 2024 [23] | Yes | Yes | Yes | Yes | Yes |
| Mourão et al. 2009 [24] | Yes | Yes | Yes | No | No |
| Naveca et al. 2018 [25] | No | Yes | Yes | No | No |
| Silva-Caso et al. 2019 [26] | Yes | Yes | Yes | Yes | Yes |
| Vasconcelos et al. 2009 [27] | Yes | Yes | Yes | Yes | Yes |
| Watts et al. 2022 [8] | Yes | Yes | Yes | Yes | Yes |

Question 1: Does the patient(s) represent(s) the whole experience of the investigator (center)?

Question 2: Was the exposure adequately ascertained?

Question 3: Was the outcome adequately ascertained?

Question 4: Was follow-up long enough for outcomes to occur?

Question 5: Is the case(s) described with sufficient detail to allow other investigators to replicate the research or to allow practitioners make inferences related to their own practice?

substantial heterogeneity. The publication bias was qualitatively assessed by visualizing the studies with a funnel plot and Egger's test.

# Results

## Study selection and characteristics

The initial search across four electronic databases yielded a total of 574 potentially relevant articles. Following the removal of duplicates and subsequent screening based on titles, abstracts, and full texts, 15 studies [8,14–27] involving 806 OROV-infected patients were eligible for inclusion (Fig 1 and S1 Data). Among the included studies, there were 12 cohort studies, two case series, and one cross-sectional study. All studies were from Latin America, with a maximum of eight from Brazil, five from Peru, and one each from Colombia and French Guiana. Detailed information is presented in Table 1. The sampling dates of the studies were broadly distributed, covering a range that extended from 1999 to 2023.

## Quality assessment

The quality assessment results of all included studies are shown in Table 2. Only one study was of poor quality, all the remaining studies were judged to be of good quality, and these patients appeared to be representative of the full range of investigator experiences, where exposures and outcomes were adequately determined, and where follow-up was long enough.

## General symptoms

All studies included in this systematic review reported general symptoms of fever and headache. Other general symptoms reported included myalgia (14 studies), arthralgia (14 studies),

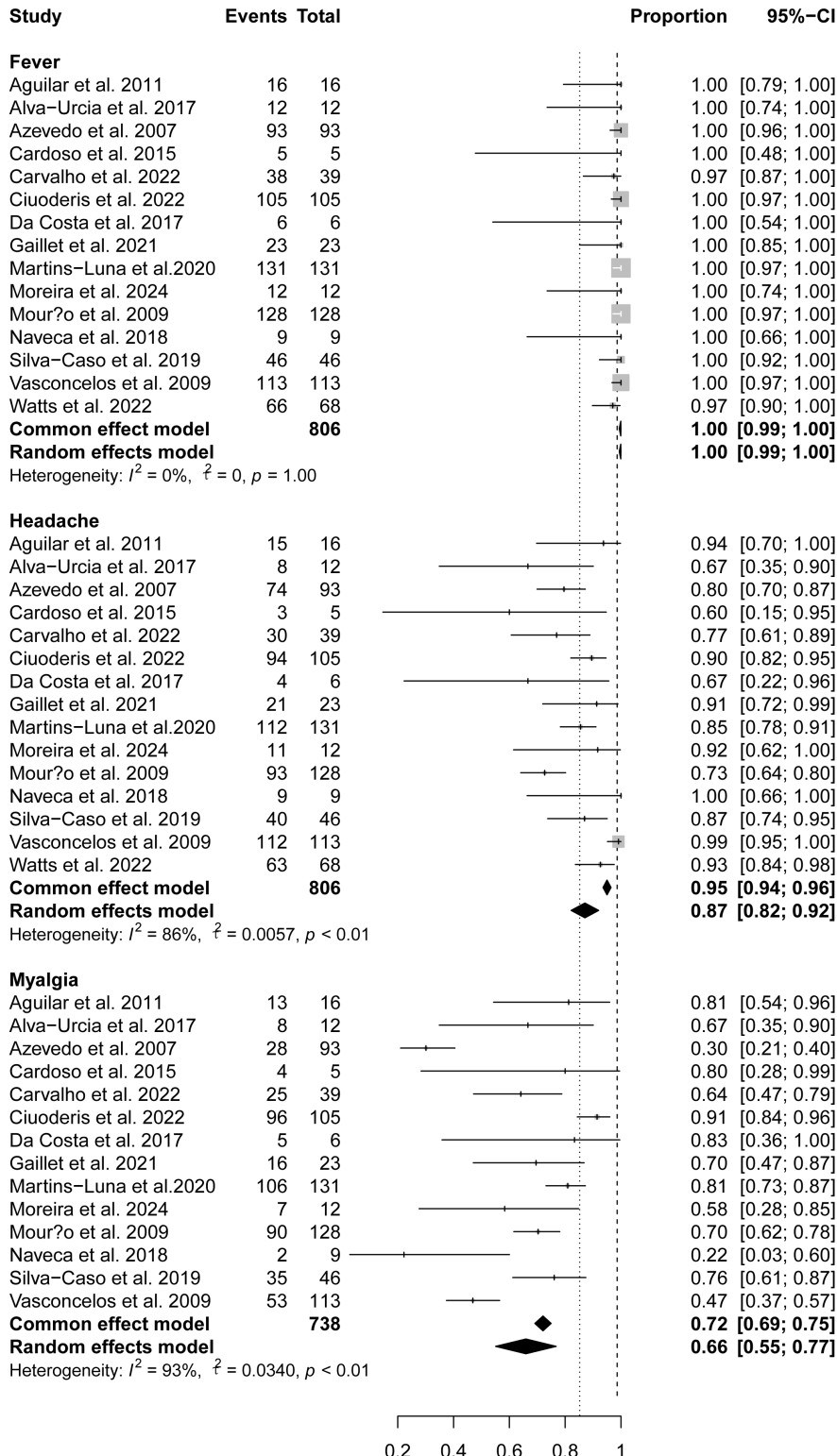

**Fig 2. Pooled prevalence of general symptoms in OROV-infected patients.**

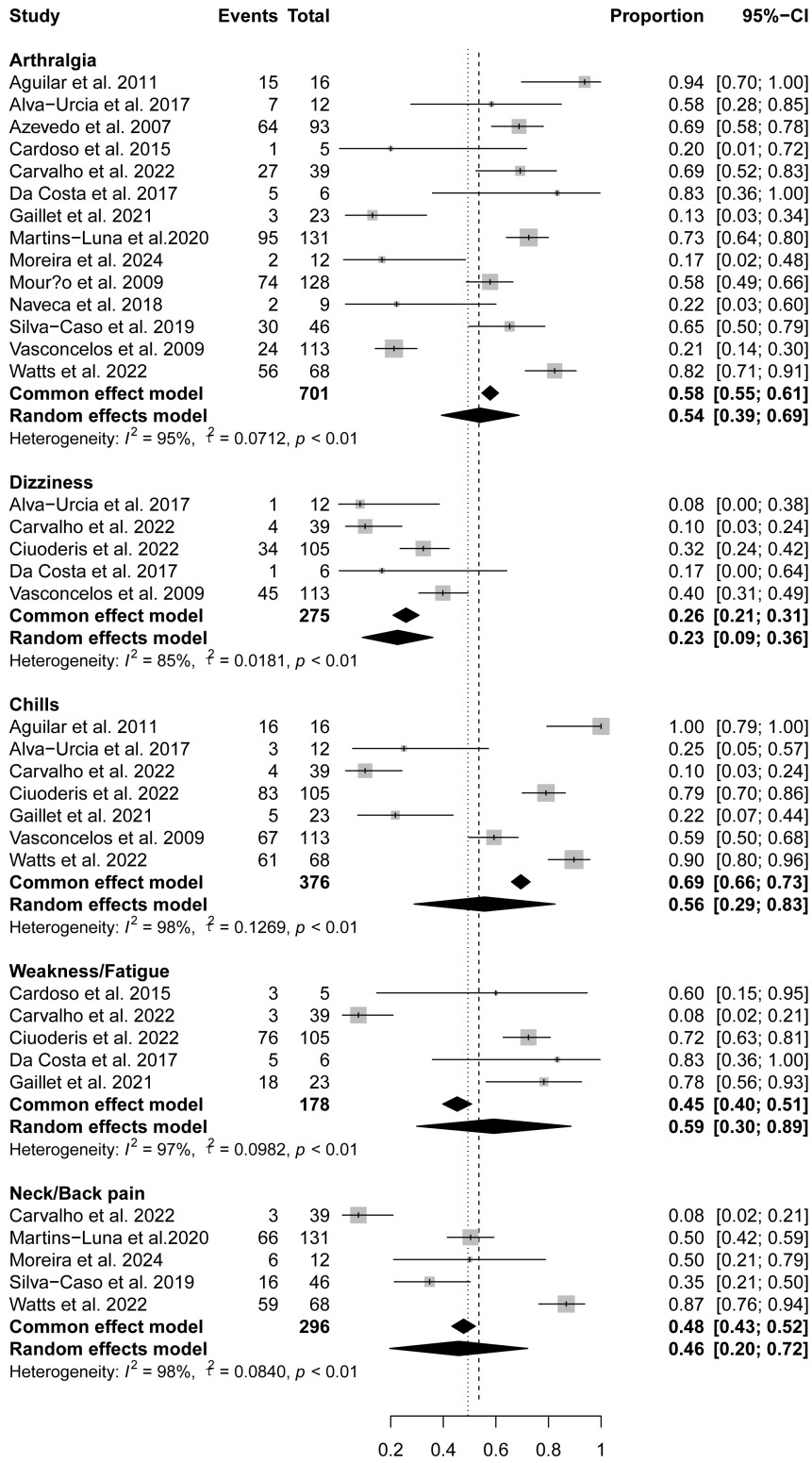

**Fig 3. Pooled prevalence of general symptoms in OROV-infected patients.**

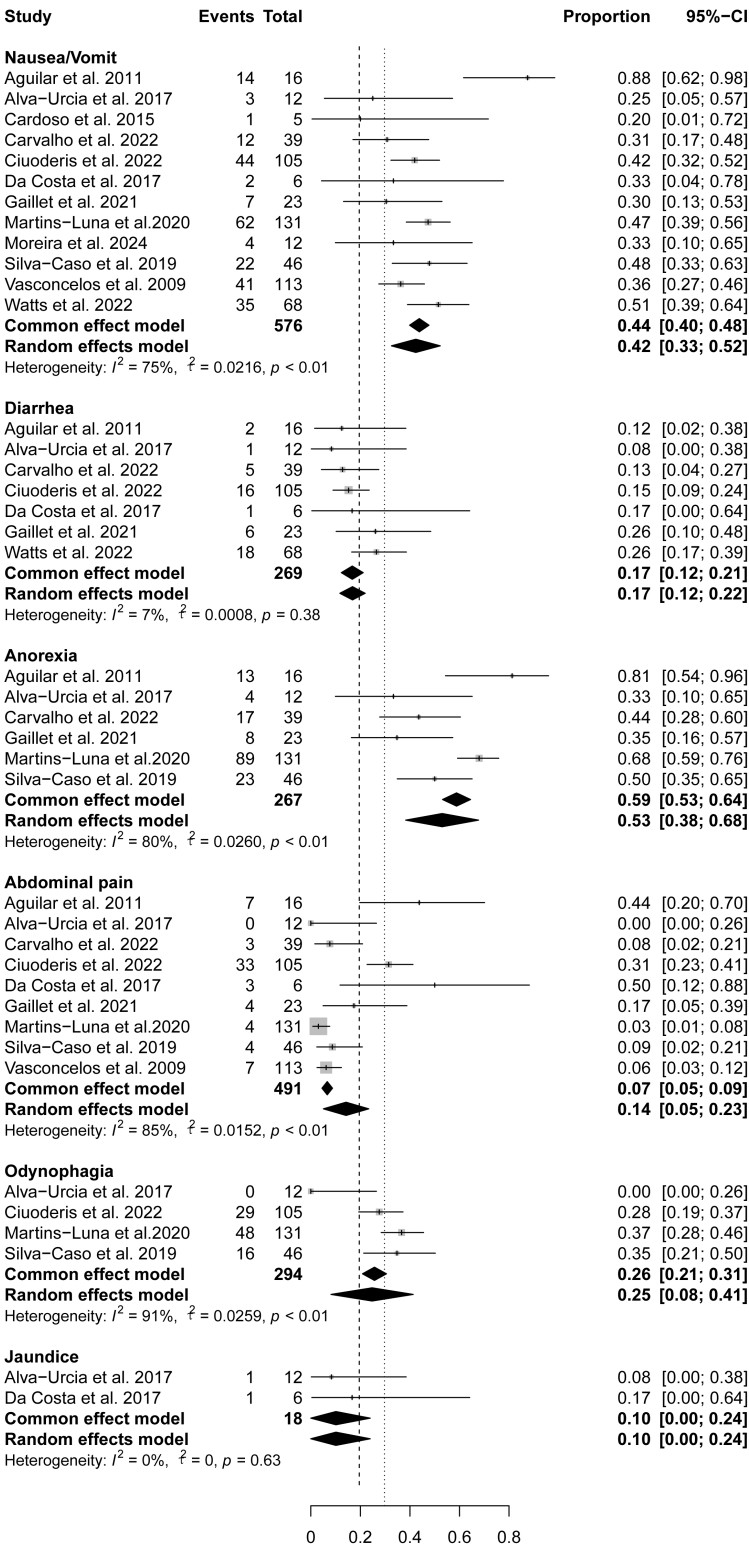

**Fig 4. Pooled prevalence of gastrointestinal symptoms in OROV-infected patients.**

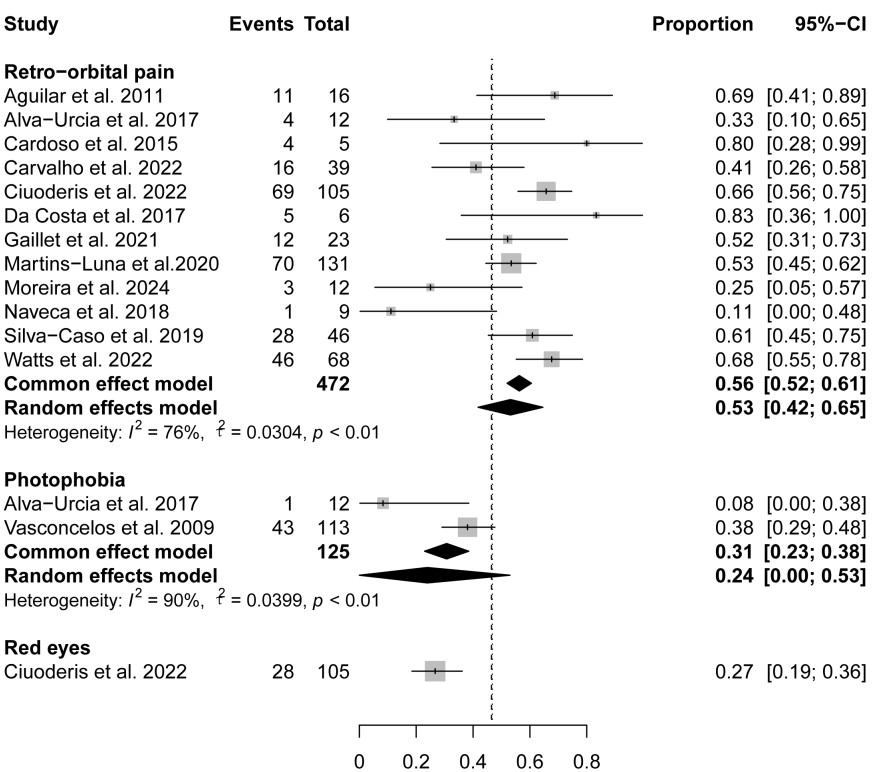

**Fig 5. Pooled prevalence of ocular symptoms in OROV-infected patients.**

chills (seven studies), dizziness (five studies), weakness/fatigue (five studies), and neck/back pain (five studies). The overall pooled prevalence for fever, headache, myalgia, arthralgia, chills, dizziness, weakness/fatigue, and neck/back pain were 100% (95% confidence interval [CI] 99%-100%), 95% (95% CI 94%-96%), 72%(95% CI 69%-75%), 58% (95% CI 55%-61%), 26%(95% CI 21%-31%), 69%(95% CI 66%-73%), 45%(95% CI 40%-51%), 48%(95% CI 43%-52%), respectively (Figs 2 and 3). All general symptoms but fever ($I^2$ = 0%) were heterogeneous among studies, with $I^2$ ranging from 85%–98%.

## Gastrointestinal symptoms

Gastrointestinal symptoms in patients with OROV infection included in this study included nausea/vomit (12 studies), abdominal pain (nine studies), diarrhea (seven studies), anorexia (six studies), odynophagia (four studies), and jaundice (two studies). The overall pooled prevalence of anorexia was the highest (59%, 95% CI 53%-64%), followed by nausea/vomit (44% 95% CI 40%-48%), odynophagia (26%, 95% CI 21%-31%), diarrhea (17%, 95% CI 12%-21%), jaundice(10%, 95% CI 0%-24%), and abdominal pain(7%, 95% CI 5%-9%) (Fig 4). Aside from diarrhea and jaundice, other gastrointestinal symptoms exhibited heterogeneity across studies, ranging from 75% to 91%.

## Ocular symptoms

In all studies included, ocular involvement was manifested as retro-orbital pain (13 studies), photophobia (two studies), and redness (one studies). Among them, retro-orbital pain was the most common (56%, 95% CI 52%-61%), followed by photophobia (31%, 95% CI 23%-38%) and red eyes (27%, 95% CI 19%-36%) (Fig 5).

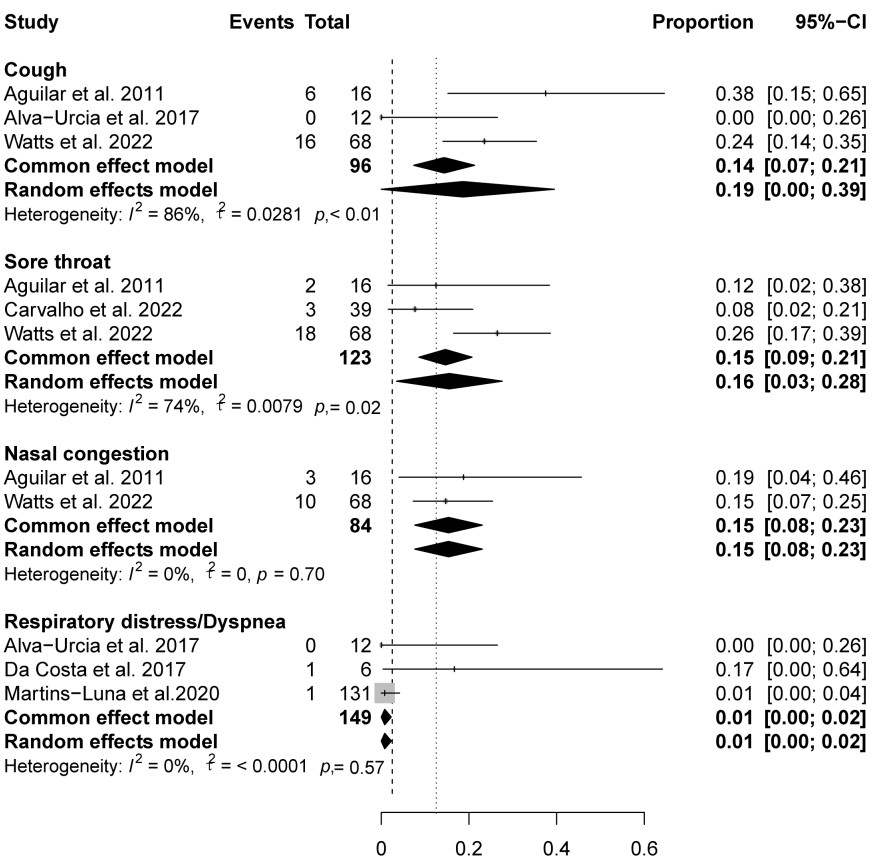

**Fig 6. Pooled prevalence of respiratory symptoms in OROV-infected patients.**

## Respiratory symptoms

Patients infected with OROV primarily present with four respiratory symptoms, typically manifesting as cough (14%, 95% CI 7%-21%), sore throat (15%, 95% CI 9%-21%), and nasal congestion (15%, 95% CI 8%-23%), while the symptom of respiratory distress/dyspnea is less common (1%, 95% CI 0%-2%) (Fig 6).

## Skin-related symptoms

OROV infection also caused skin-related symptoms, including rash, pruritus, and pallor, with the highest overall pooled prevalence of pallor (28%, 95% CI 12%-44%), followed by pruritus (12%, 95% CI 3%-21%) and rash (11%, 95% CI 9%-14%) (Fig 7).

## Discussion

This systematic review is the first to explore pooled prevalence of clinical features and complications of OROV infection.

Fever and headache are common symptoms in individuals infected with OROV, with overall prevalence rates of 100% and 95% respectively, compared to a higher prevalence observed in dengue fever infections [28]. However, this difference should be interpreted cautiously, as the current scarcity of epidemiological monitoring studies on OROV, with many research efforts focused solely on patients exhibiting clinical fever but testing negative for dengue, may

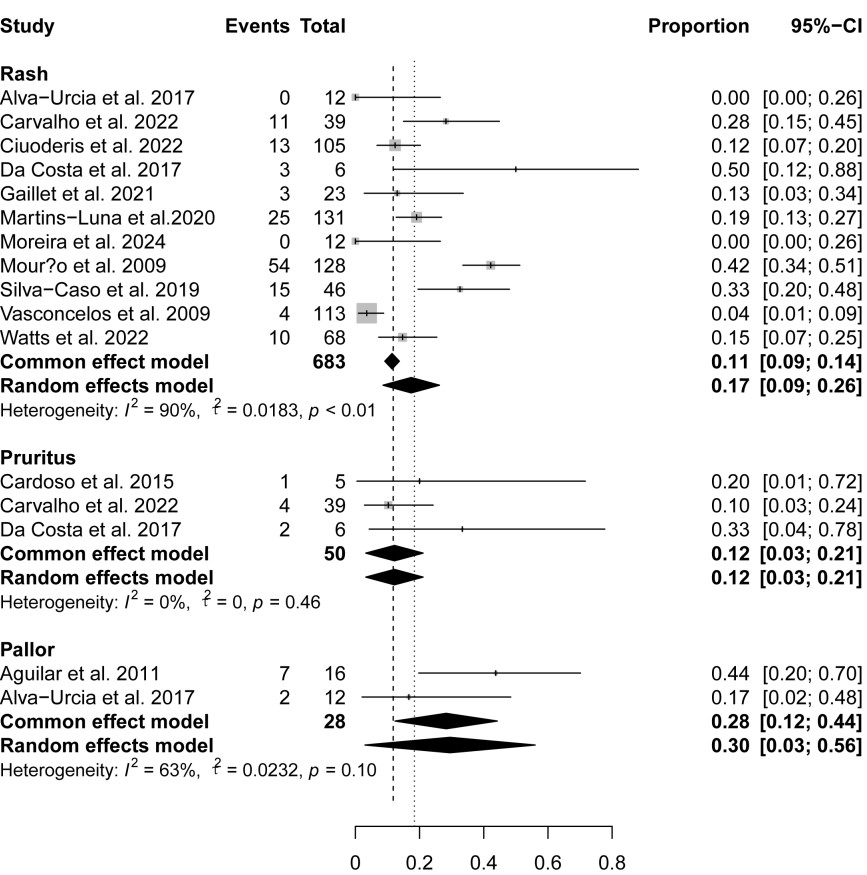

**Fig 7. Pooled prevalence of skin-related symptoms in OROV-infected patients.**

lead to an overestimation of these symptoms' prevalence due to selection bias. The overall pooled prevalence of myalgia, joint pain and chills is comparable to dengue and Zika infections [28,29].

OROV infections are commonly associated with gastrointestinal symptoms, including anorexia, nausea/vomit, odynophagia, diarrhea, and abdominal pain. These symptoms show similarities with those observed in dengue fever infections [29]. Moreover, the observation of jaundice in two of our studies indicates that the viral infection may lead to hemolysis or liver injury in certain cases. The prominence of ocular involvement, particularly retro-orbital pain, photophobia, and red eyes. Ophthalmologic evaluations should be integrated into the management protocol for suspected OROV cases to ensure timely intervention and reduce the risk of ocular complications. Respiratory symptoms, including cough, sore throat, and nasal congestion, and occur in a minority of OROV patients, with prevalences ranging from 14% to 15%. The infrequent occurrence of respiratory distress (1%) suggests that severe pulmonary complications are uncommon, distinguishing OROV from respiratory viruses like influenza and SARS-CoV-2 [30,31]. Notably, rash occurs in only 11% of OROV infections, significantly lower than in dengue fever and Zika virus infections [29,32].

This review has several limitations. The main limitation is that the included studies were mostly retrospective studies and lacked high-quality data. Additionally, patients who were asymptomatic or had mild cases and who did not require hospitalization were not accounted

for due to publication bias. The systematic review may also have missed studies that were not indexed or published in local journals, especially in languages other than English.

## Conclusion

In conclusion, the multi-systemic nature of OROV infections, as revealed by this systematic review, calls for a heightened clinical awareness and a comprehensive approach to patient evaluation. By recognizing the breadth of symptoms associated with OROV, healthcare providers can more accurately diagnose and manage this infection, thereby contributing to better health outcomes and facilitating the development of targeted therapeutic interventions.

## Supporting information

**S1 Data. Original data of OROV-infected patients included in the study.**
(XLSX)

**S1 Checklist. PRISMA 2020 Checklist.**
(DOCX)

## Acknowledgements

This study was non-financially supported in by Baise Talent Highland (No. 2020-3-2), Building Projects of Guangxi Bagui Scholars (No. Guirencaiban [2024]-29), Building Projects of the Key Laboratory of Molecular Pathology in Tumor of Guangxi Higher Education Institutes (No. Guijiaokeyan [2022]-10), Building Projects of the Key Laboratory of Molecular Pathology in Tumor of Baise (No. 2022), and Clinical Key Specialty Building Project (For Pathology) of Guangxi (No. Guiweiyifa [2022]-21).

## Author contributions

**Conceptualization:** Zhilin Wang, Linzhu Huang, Yulei Li.

**Formal analysis:** Zhilin Wang, Liwei Huang, Yulei Li.

**Funding acquisition:** Demin Cao, Yulei Li.

**Investigation:** Xinyu Zhang, Xinyue Zhang.

**Methodology:** Zhilin Wang, Linzhu Huang, Xinyu Zhang, Liwei Huang, Xiaoying Zhu.

**Project administration:** Xinyu Zhang.

**Software:** Xinyu Zhang, Liwei Huang, Xiaoying Zhu.

**Supervision:** Xiaoying Zhu, Xidai Long, Demin Cao, Yulei Li.

**Validation:** Xinyue Zhang, Xiaoying Zhu, Yulei Li.

**Visualization:** Xinyu Zhang, Xinyue Zhang, Demin Cao.

**Writing – original draft:** Xinyue Zhang, Xidai Long, Demin Cao, Yulei Li.

**Writing – review & editing:** Zhilin Wang, Linzhu Huang, Xidai Long, Demin Cao, Yulei Li.

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
