## [Decision Letter · Decision Letter 0]

28 Jan 2025

PNTD-D-24-01266Clinical presentation of Oropouche virus infection: a systematic review and meta-analysisPLOS Neglected Tropical Diseases  Dear Dr. Li, Thank you for submitting your manuscript to PLOS Neglected Tropical Diseases. After careful consideration, we feel that it has merit but does not fully meet PLOS Neglected Tropical Diseases's publication criteria as it currently stands. Therefore, we invite you to submit a revised version of the manuscript that addresses the points raised during the review process. Please submit your revised manuscript within 30 days Mar 29 2025 11:59PM. If you will need more time than this to complete your revisions, please reply to this message or contact the journal office at plosntds@plos.org. Please include the following items when submitting your revised manuscript: * A rebuttal letter that responds to each point raised by the editor and reviewer(s). You should upload this letter as a separate file labeled 'Response to Reviewers '. This file does not need to include responses to any formatting updates and technical items listed in the 'Journal Requirements' section below. * A marked-up copy of your manuscript that highlights changes made to the original version. You should upload this as a separate file labeled 'Revised Manuscript with Track Changes '. * An unmarked version of your revised paper without tracked changes. You should upload this as a separate file labeled 'Manuscript '. If you would like to make changes to your financial disclosure, competing interests statement, or data availability statement, please make these updates within the submission form at the time of resubmission. Guidelines for resubmitting your figure files are available below the reviewer comments at the end of this letter. We look forward to receiving your revised manuscript. Kind regards, Adly M.M. Abd-Alla, Prof asso.Academic EditorPLOS Neglected Tropical Diseases

Elvina Viennet

Section Editor

Shaden Kamhawi

co-Editor-in-Chief

Paul Brindley

co-Editor-in-Chief

**Journal Requirements:**

At this stage, the following Authors/Authors require contributions: Yulei Li, Mengxi Gu, Xinyue Zhang, Xiaoying Zhu, Demin Cao, and Xidai Long. Please ensure that the full contributions of each author are acknowledged in the "Add/Edit/Remove Authors" section of our submission form.

5) We note that your Data Availability Statement is currently as follows: "The original contributions presented in the study are included in the article/supplementary material. Further inquiries can be directed to the corresponding author.". Please confirm at this time whether or not your submission contains all raw data required to replicate the results of your study. Authors must share the “minimal data set” for their submission. PLOS defines the minimal data set to consist of the data required to replicate all study findings reported in the article, as well as related metadata and methods (https://journals.plos.org/plosone/s/data-availability#loc-minimal-data-set-definition).

- The points extracted from images for analysis..

**Reviewers' comments:** Reviewer's Responses to Questions

**Key Review Criteria Required for Acceptance?**

**Methods**

-Are the objectives of the study clearly articulated with a clear testable hypothesis stated?

-Is the study design appropriate to address the stated objectives?

-Is the population clearly described and appropriate for the hypothesis being tested?

-Is the sample size sufficient to ensure adequate power to address the hypothesis being tested?

-Were correct statistical analysis used to support conclusions?

-Are there concerns about ethical or regulatory requirements being met?

Reviewer #1: The objectives are clear and they have been achieved. Some comments:

"OROV is an acute febriel disease..." should be replaced by "OROV causes an acute febriel disease..." as OROV is the virus, nor the disease.

There is not and evident lack of literature; there is an evident lack of systematic review of the scattered data. So, better reformulate the sentence... "To date, the comprehensive.... as ther is an evident lack of systematic reviews."

Why does this lower limit of studies of less than five cases make these studies ineligible? They keep adding cases to the final "n".

What meaning do the numbers 5 and 8 have next to Peru and Brazil, respectively, in Table 1? No information about this appears in the table key.

The phrase...."The prominence of ocular involvement, particularly retro-orbital pain, photophobia, and red eyes." lacks a clear meaning for me. It seems that the verb is missing.

Remember that most taxonomic names are in italics. Check references 10, 16.

Reviewer #2: Are the objectives of the study clearly articulated with a clear testable hypothesis stated?

Yes, the objectives of the study are clearly articulated. The study aims to systematically review and meta-analyze the clinical presentation of Oropouche virus (OROV) infection. The hypothesis is implied rather than explicitly stated, but the study seeks to provide a comprehensive understanding of the clinical manifestations of OROV infection, which is a clear and testable objective.

Is the study design appropriate to address the stated objectives?

Yes, the study design is appropriate. The authors conducted a systematic review and meta-analysis, which is suitable for summarizing the clinical manifestations of OROV infection across multiple studies. The inclusion and exclusion criteria are well-defined, and the search strategy is comprehensive.

Is the population clearly described and appropriate for the hypothesis being tested?

Yes, the population is clearly described. The study includes 806 patients from 15 studies, all of whom were confirmed to have OROV infection via PCR or ELISA. The population is appropriate for the hypothesis being tested, as it focuses on individuals with confirmed OROV infection.

Is the sample size sufficient to ensure adequate power to address the hypothesis being tested?

Yes, the sample size of 806 patients from 15 studies is sufficient for a meta-analysis. The authors have included a reasonable number of studies and patients to ensure adequate power to address the hypothesis.

Were correct statistical analyses used to support conclusions?

Yes, the statistical analyses appear to be appropriate. The authors used a fixed-effect model for meta-analysis and assessed heterogeneity using the I² statistic. They also used forest plots and funnel plots to visualize the data and assess publication bias. The methods are well-described and appropriate for the type of data being analyzed.

Are there concerns about ethical or regulatory requirements being met?

No, there are no apparent concerns about ethical or regulatory requirements. The study is a systematic review and meta-analysis of previously published studies, so it does not involve direct human or animal subjects. The authors have declared no competing interests and no specific funding for this work.

**Results**

-Does the analysis presented match the analysis plan?

-Are the results clearly and completely presented?

-Are the figures (Tables, Images) of sufficient quality for clarity?

Reviewer #1: Appropriate and sufficient. If the quality and disaggregation of the clinical data in the different studies allowed it, a review of the clinical presentations by age or sex would have been interesting to read.

Reviewer #2: Does the analysis presented match the analysis plan?

Yes, the analysis presented matches the analysis plan outlined in the methods section. The authors conducted a meta-analysis of clinical symptoms and presented the results in a clear and structured manner.

Are the results clearly and completely presented?

Yes, the results are clearly and completely presented. The authors provide detailed tables and figures summarizing the prevalence of various symptoms, including general, gastrointestinal, ocular, respiratory, and skin-related symptoms. The results are well-organized and easy to follow.

Are the figures (Tables, Images) of sufficient quality for clarity?

Yes, the figures and tables are of sufficient quality for clarity. The forest plots and tables are well-constructed and provide a clear visual representation of the data. The figures are appropriately labeled and easy to interpret.

**Conclusions**

-Are the conclusions supported by the data presented?

-Are the limitations of analysis clearly described?

-Do the authors discuss how these data can be helpful to advance our understanding of the topic under study?

-Is public health relevance addressed?

Reviewer #1: The conclusion are supported by the data presented. Limitations of the analysis are clearly described but not completely assessed. Taking into account that the circulation of this virus basically occurs in Latin America, how many studies in Spanish or Portuguese/Brazilian have been not considered and therefore discarded? How many cases? What impact does this selection have on the final result?

The discussion is sufficient and obviously it has public health relevance for the professionals in the field.

Reviewer #2: Are the conclusions supported by the data presented?

Yes, the conclusions are supported by the data presented. The authors conclude that OROV infection presents with a wide range of symptoms, including fever, headache, gastrointestinal disturbances, ocular symptoms, and respiratory symptoms. These conclusions are well-supported by the meta-analysis results.

Are the limitations of analysis clearly described?

Yes, the limitations of the analysis are clearly described. The authors acknowledge that most of the included studies were retrospective and that there may be publication bias due to the exclusion of asymptomatic or mild cases. They also note the potential for selection bias and the lack of high-quality data in some studies.

Do the authors discuss how these data can be helpful to advance our understanding of the topic under study?

Yes, the authors discuss the public health relevance of their findings. They emphasize the need for heightened clinical awareness and a comprehensive approach to patient evaluation, which can help in the accurate diagnosis and management of OROV infection. This discussion is relevant and adds value to the study.

Is public health relevance addressed?

Yes, the public health relevance is addressed. The authors highlight the increasing incidence and geographic spread of OROV infections and the need for targeted public health interventions. They also discuss the importance of integrating ophthalmologic evaluations into the management protocol for suspected OROV cases.

**Editorial and Data Presentation Modifications?**

Reviewer #1: This reviewer believes that a new reading of the text and improving the constructions of some sentences would improve the manuscript.

Reviewer #2: The authors should consider explicitly stating the hypothesis in the introduction to make the study's objectives even clearer.

The discussion could be strengthened by comparing the findings with other arboviral infections (e.g., dengue, Zika) in more detail, as this would provide additional context for the clinical significance of OROV symptoms.

The authors should ensure that all abbreviations are defined upon first use (e.g., "OROV" is defined, but other abbreviations like "PCR" and "ELISA" are not).

**Summary and General Comments**

Reviewer #1: This is a systematic review of the clinical symptoms of Oropouche virus infection based on previous work. Systematizes dispersed data and organizes it. Useful for case management and public health.

Reviewer #2: Strengths:

The study is well-designed and addresses an important gap in the literature regarding the clinical presentation of OROV infection.

The meta-analysis is conducted rigorously, with appropriate statistical methods and clear presentation of results.

The study has significant public health relevance, particularly in regions where OROV is endemic or emerging.

Weaknesses:

The study is limited by the retrospective nature of most included studies and potential publication bias, as acknowledged by the authors.

The discussion could be expanded to provide more context on how OROV symptoms compare to other arboviral infections, which would enhance the study's clinical relevance.

Novelty and Significance:

This is the first systematic review and meta-analysis to comprehensively summarize the clinical manifestations of OROV infection. The findings are novel and provide a foundational understanding that can guide future research and public health interventions.

General Execution and Scholarship:

The study is well-executed and demonstrates a high level of scholarship. The authors have followed appropriate guidelines for systematic reviews and meta-analyses, and the manuscript is well-written and organized.

PLOS authors have the option to publish the peer review history of their article (what does this mean? ). If published, this will include your full peer review and any attached files.

**Do you want your identity to be public for this peer review?** For information about this choice, including consent withdrawal, please see our Privacy Policy .

Reviewer #1: No

Reviewer #2: No

**Figure resubmission:**

 **Reproducibility:** To enhance the reproducibility of your results, we recommend that authors of applicable studies deposit laboratory protocols in protocols.io, where a protocol can be assigned its own identifier (DOI) such that it can be cited independently in the future. Additionally, PLOS ONE offers an option to publish peer-reviewed clinical study protocols. Read more information on sharing protocols at https://plos.org/protocols?utm_medium=editorial-email&utm_source=authorletters&utm_campaign=protocols

---

## [Editor Report · Decision Letter 1]

4 Mar 2025

Dear Dr Li,

We are pleased to inform you that your manuscript 'Clinical presentation of Oropouche virus infection: a systematic review and meta-analysis' has been provisionally accepted for publication in PLOS Neglected Tropical Diseases.

Best regards,

Adly M.M. Abd-Alla, Prof asso.

Academic Editor

Elvina Viennet

Section Editor

Shaden Kamhawi

co-Editor-in-Chief

Paul Brindley

co-Editor-in-Chief

---

## [Editor Report · Acceptance letter]

Dear Dr Li,

We are delighted to inform you that your manuscript, "Clinical presentation of Oropouche virus infection: a systematic review and meta-analysis," has been formally accepted for publication in PLOS Neglected Tropical Diseases.

Best regards,

Shaden Kamhawi

co-Editor-in-Chief

Paul Brindley

co-Editor-in-Chief
